# Apoptosis in idiopathic inflammatory myopathies with partial invasion; a role for CD8+ cytotoxic T cells?

Olof Danielsson[1]*, Bo Häggqvist[1], Liv Gröntoft[1], Karin Öllinger[2], Jan Ernerudh[3]

**1** Division of Neurology, Department of Biomedical and Clinical Sciences, Faculty of Medicine and Health Sciences, Linköping University, Linköping, Sweden, **2** Division of Experimental Pathology, Department of Biomedical and Clinical Sciences, Linköping University, Linköping, Sweden, **3** Division of Clinical Immunology and Transfusion Medicine, Department of Biomedical and Clinical Sciences, Linköping University, Linköping, Sweden

* olof.danielsson@regionostergotland.se

**Data Availability Statement:** All relevant data are within the manuscript and its Supporting Information files.

## Abstract

Polymyositis and inclusion body myositis are idiopathic inflammatory myopathies, with a pathology characterized by partial invasion of non-necrotic muscle fibres by CD8+ cytotoxic T-cells, leading to fibre degeneration. Although the main effector pathway of CD8+ T-cells is to induce apoptosis of target cells, it has remained unclear if apoptosis occurs in these diseases, and if so, if it is mediated by CD8+ T-cells. In consecutive biopsy sections from 10 patients with partial invasion, muscle fibres and inflammatory cells were assessed by immunohistochemistry and apoptotic nuclei by the TUNEL assay. Analysis of muscle fibre morphology, staining pattern and quantification were performed on digital images, and they were compared with biopsies from 10 dermatomyositis patients and 10 controls without muscle disease. Apoptotic myonuclei were found in muscle with partial invasion, but not in the invaded fibres. Fibres with TUNEL positive nuclei were surrounded by CD8+ T-cells, granzyme B+ cells and macrophages, but lacked FAS receptor expression. In contrast, apoptotic myonuclei were rare in dermatomyositis and absent in controls. The findings confirm that apoptosis occurs in idiopathic inflammatory myopathies and support that it is mediated by CD8+ cytotoxic T- cells, acting in parallel to the process of partial invasion.

## Introduction

Idiopathic inflammatory myopathies (IIM) constitute a heterogeneous group of diseases with a presumed autoimmune pathogenesis [1]. They are commonly classified on the basis of clinicopathologic features [2], although variations due to overlapping or unspecific clinical and pathological findings sometimes make diagnosis and classification a challenge [3–5]. Key pathologic findings, distinguishing the inflammatory myopathies, are the localization of the inflammatory infiltrates and the different types of muscle fibre degeneration [5, 6]. Muscle biopsies from patients with polymyositis (PM) and inclusion body myositis (IBM) characteristically show invasions of major histocompatibility complex class I—(MHC I) expressing muscle

**Funding:** OD received a finacial grant from Region Östergötland. Grant no: LIO-535621. The funders had no role in study design, data collection and analysis, decision to publish, or preparation of the manuscript

**Competing interests:** The authors have declared that no competing interests exist.

fibres by an inflammatory infiltrate, dominated by CD8[+] cytotoxic T cells [6]. This scenario is commonly referred to as a partial invasion, and is included as a classification criterion for PM, intended for research studies [7]. The invaded fibre does not show classical signs of necrosis, but rather a gradual disintegration and displacement by inflammatory cells [8]. This type of cell degeneration is not found in other tissues, and is rarely seen in other muscle diseases [6]. In dermatomyositis (DM), considered as a humorally mediated microangiopathy affecting muscle and skin, the most characteristic pathologic muscle findings are perivascular inflammatory infiltrates and perifascicular atrophy [2]. Another widely accepted pathology based subgroup of IIM is immune-mediated necrotizing myopathy (IMNM), also called necrotizing autoimmune myopathy (NAM) [7], characterized by muscle fibre necrosis and the relative absence of inflammatory cells. In addition, there is an ongoing discussion about further myopathological subdivisions of these diseases [5, 9]. However, partial invasion is a feature that is common to both IBM and PM (strictly defined), and extensive sectioning of muscle tissue shows that partial invasion is sometimes encountered also in overlap syndromes, and, rarely, in classical DM [4, 6]. This raises the question whether there are further shared pathogenic processes, associated with partial invasion.

The general effector mechanism mediated by cytotoxic CD8[+] T-cells is to induce apoptosis, either by means of granzyme/perforin secretion or by binding of the FAS-ligand (FAS-L) to FAS receptors (hereafter called FAS) of target cells [10–13]. Expression of granzyme B and perforin in inflammatory cells [14, 15], as well as the expression of FAS in muscle fibres [16–18], has been reported in inflammatory myopathies, but signs of apoptotic fibre nuclei have rarely been observed in fibres affected by partial invasion [19, 20]. Overall, signs of apoptosis in IIM muscle fibres have, despite reports of its presence [20, 21], been considered rare, and unlikely of importance for the pathogenesis [20, 22]. Absence of apoptosis has also been reported for the inflammatory cells in IIM, which has been interpreted as an inability to efficiently terminate the inflammatory reaction, possibly contributing to the chronicity of these diseases [22, 23].

The pathogenic role of CD68[+] macrophages, the second most frequent cell type in the invading infiltrate [6], is still unclear. Notably, no ultrastructural signs of phagocytosis have been noted [8], and the macrophages express markers of myeloid dendritic cells [24], stressing their antigen presenting potential. Macrophages within the invading infiltrate have further been suggested to contribute to the distinctive pathology by producing inflammatory molecules [25]. Macrophages expressing the scavenger receptor CD163 (type 2 macrophages) have been attributed anti-inflammatory properties, mediated by IL-10 [26], and a regenerating role in muscle [27]. On the other hand, a correlation between disease activity in IIM muscle and CD163[+] macrophages has been reported [28, 29], but their relative number in relation to CD68[+] macrophages has not been investigated.

Considering the main cytotoxic effector functions of the CD8[+] T-cells, characteristically present in IBM and polymyositis, several studies have identified factors with potential to protect muscle from apoptosis [16, 18, 30, 31]. The fact that muscle fibres do not normally express MHC I [32], but constitutionally HLA-G [33], and the rather uncommon occurrence of inflammatory muscle diseases, indicate a relative protected immunologic status of striated muscle. In line with this, we earlier reported a constitutional expression of Bcl-2, a major anti-apoptotic protein, in muscle from healthy controls, but lower Bcl-2 expression in PM and Duchenne muscular dystrophy [18]. Bcl-2 has potential to protect against granzyme-mediated apoptosis [34], but its precise role in muscle is unknown. Heat shock proteins (HSP) have cytoprotective functions that may be of importance [35], considering that the sarcoplasmatic reticulum stress, observed in IIM, may lead to apoptosis [36]. HSP70 has recently been shown to protect against apoptosis in cardiomyocytes [37]. Moreover, HSP70 has been detected in inflammatory myopathies [38, 39] and, also in this scenario, been attributed a protective function [40].

The aim of the study was to investigate if apoptotic myonuclei are present in muscle in the specific subgroup of IIM with partial invasion, and to detect the presence of key molecules of the apoptotic-mediating or -protective pathways in the affected and in adjacent fibres. The CD8[+] T-cell-mediated immune effector mechanism and frequency of type 2 macrophages were evaluated.

## Methods

### Patients and biopsies

All biopsies were taken from the anterior tibial muscle for diagnostic purposes. We included stored tissue samples, obtained 1997–2002, with confirmed partial invasion from 10 patients (median age 62 years, range 32–75) with inflammatory myopathies (Table 1), which in a previous study [4] had been classified according to the Amato/European neuromuscular centre (ENMC) classification [7]. These samples were compared with two control groups, comprising 10 patients in each group. As an IIM control group, biopsies from patients with DM were selected; the reason for this choice was that this disease has another type of, more or less pathognomonic, pathological findings [7]. The DM cases (median age 52 years, range17-78) were consecutive biopsies investigated in our lab, where the available clinical and laboratory data allowed the classification of at least probable DM, according to the Amato/ENMC. Eight cases had a definite and two a probable DM according to this classification, and all cases had a definite DM according to the Bohan and Peter. The other control group comprised consecutive cases (median age 46 years, range 39–59), where neuromuscular disease had been excluded, based on biopsy and clinical examination. The cases in these latter two groups were included during April 2015. For demographic details of all cases, see S1 Dataset.

### Sectioning method and selection of cases with partial invasion

In the earlier study [4], muscle biopsies from 36 patients were shown to contain at least one partial invasion. In order to detect and verify strict biopsy criteria stated by Amato/ENMC [7], 35 consecutive cryosections had been made from each biopsy, and every 7[th] section (6 μm)

**Table 1. Demography, diagnosis* and results** (number of partial invasions, TUNEL stained myonuclei, and necrotic fibres) of patients with partial invasion***.**

| Case | Age (yrs) | Sex | Bohan & Peter | Amato/ENMC | Partial inv. | Necrotic fibres | TUNEL+ fibres | TUNEL+ infl. cells |
|------|-----------|-----|---------------|------------|--------------|-----------------|---------------|---------------------|
| 1 | 66 | m | IBM def. | IBM def. | 1 | 0 | 1 | 0 |
| 2 | 62 | f | Overlap IIM pr. | non IIM# | 3 | 1 | 0 | 1 |
| 3 | 58 | f | Overlap IIM def. | Poss. DM s. D. | 4 | 0 | 1 | 0 |
| 4 | 32 | f | PM def. | PM def. | 1 | 1 | 0 | 1 |
| 5 | 66 | f | Overlap IIM pr. | PM def. | 3 | 0 | 1 | 7 |
| 6 | 53 | f | PM poss. | non IIM# | 1 | 0 | 0 | 1 |
| 7 | 60 | m | IBM def. | IBM def. | 1 | 2 | 0 | 3 |
| 8 | 73 | m | IBM poss. | IBM poss. | 3 | 1 | 0 | 2 |
| 9 | 70 | m | IBM def. | IBM def. | 1 | 0 | 2 | 3 |
| 10 | 75 | f | PM pr. | PM def. | 2 | 0 | 3 | 3 |

*The diagnoses were according to the classifications of Bohan & Peter[41] and Amato/ European Neuromuscular Centre (ENMC) [7]. The IBM diagnosis was in both classifications according to Griggs [42].

**The investigated muscle section area was 1.37 mm$^2$.

*** The creatine kinase was elevated in all patients. # These two patients did not have a detectable muscle weakness and could thus not be classified according to Amato/ ENMC. m: male, f: female, def.: definitive, pr.: probable, poss.: possible, IIM: idiopathic inflammatory myopathy, non-IIM: patients without weakness does not qualify as IIM according to Amato/ENMC, IBM: inclusion body myositis, PM: polymyositis, poss. DM s. D: possible dermatomyositis sine dermatitis.

stained with haematoxylin eosin (HE). Partial invasions were first identified by the morphologic appearance in the HE stained sections and then confirmed by the following three sections, stained with antibodies against MHC I, CD8 and membrane attack complex (MAC). MAC was included to add sensitivity to detect myofibre necrosis [43]. For the present study, the frozen consecutive sections (-70˚C) of the biopsies were thawed, and the HE stained sections were again studied in a Zeiss AXIO Imager light microscope. From the 36 cases, we selected those where the partial invasion were present in at least 3 HE sections (n = 10), and used the intermediate sections for immunohistochemistry and the TUNEL-assay.

## Reflections on the representativeness of the included cases

The requirement that the partial invasion had to be detectable in 3 HE stained sections, had the effect that partial invasions with a smaller size than approximately 150 μm were not included. We have not found any report voicing the possibility that partial invasions of different sizes may differ in a qualitative manner, but it cannot be excluded. We earlier showed that partial invasions is present in many types of IIM [4]. However, it is more common and more frequent in IBM, followed by PM [44]. The distribution of diagnoses of investigated pi-cases is comparable with that found in the 36 cases in our 6 year IIM cohort [4] (Table 1). With the above reservation of pi-size, the results are thus considered representative for IIM with pi, not of any defined disease subgroup.

## Immunohistochemistry

The sections were labelled with the following monoclonal antibodies: Bcl-2, CD68, CD163, FAS, Granzyme B, HSP70, Merosin 80, MyHC-fast, MyHC-slow and Spectrin (for details see Table 2). Primary antibodies were diluted in Da Vinci Green diluent (BiocareMedical, CA, USA) and incubated at room temperature in a humid chamber. Tris-buffered saline (TBS; 50 mM), pH 7.4, was used as washing buffer. Bound antibodies were detected using the Novolink Polymer Detection System (Leica Biosystems Nussloch GmbH, Germany), according to the manufacturer's protocol. Novocastra DAB enhancer was used to improve the Bcl-2 visualization. Negative

**Table 2. Antibodies and immunohistochemical procedures.**

| Antigen | Clone¤ | Manufacturer | Isotype | Incubation | Dilution | Fixation and time |
|---|---|---|---|---|---|---|
| Bcl-2 | 100 | Santa Cruz* | IgG1 | 18 h | 1:10 | Acetone, 10 min |
| CD8 | C8/144B | DAKO** | IgG1 | 30 min | 1:200 | PFA 2%, 8 min |
| CD68 | KP1 | Santa Cruz | IgG1 | 30 min | 1:400 | PFA 2%, 8 min |
| CD163 | GHI/61 | Santa Cruz | IgG1 | 30 min | 1:400 | PFA 2%, 8 min |
| FAS | B-10 | Santa Cruz | IgG1 | 18 h | 1:800 | Acetone, 10 min |
| Granzyme B | GB11 | Santa Cruz | IgG1 | 30 min | 1:1600 | PFA 2%, 8 min |
| HSP70 | C92F3A-5 | Santa Cruz | IgG1 | 30 min | 1:400 | PFA 2%, 8 min |
| Merosin 80 | MAB1922 | Millipore# | IgG1 | 30 min | 1:100 | Unfixed |
| MyHC-fast | WB-MHCf | Leica## | IgG1 | 30 min | 1:200 | Acetone, 10 min |
| MyHC-slow | WB-MHCs | Leica | IgG1 | 30 min | 1:400 | Acetone, 10 min |
| Spectrin | RBC2/3D5 | Leica | IgG2b | 30 min | 1:50 | PFA 2%, 8 min |

¤ All monoclonal antibodies were from mice and all incubations were at room temperature.

MyHC, myosin heavy chain; PFA: paraformaldehyde.

* Santa Cruz Biotechnology, Inc. Dallas TX;

** Dako Denmark A/S; # Millipore Corp, Temecula, CA; ## Leica Biosystems, Nussloch GmbH.

controls were performed by replacing the primary monoclonal antibody with IgG1 isotype control (X0931, Dako Denmark A/S) or by omitting the primary antibody.

### Detection of apoptotic nuclei

ApopTag® Plus Peroxidase *In Situ* Apoptosis Detection kit (EMD Milipore Corp., Temecula CA, USA) was used for TUNEL assay, according to the manufacturer's protocol [45]. Of three consecutive sections, the first and third were stained, using the second as negative control. In addition to application of nuclease on biopsy samples and apoptosis-positive female rodent mammary glands, supplied by the manufacturer, human heart tissue from autopsy were used as positive controls. The heart tissue, friendly supplied by Erik Edston, had been used in an earlier study to detect apoptotic cardiomyonuclei [46], where the same TUNEL method was applied as in our study.

### Complementary stains with spectrin, CD8 and granzyme B

In order to clearly discriminate, between apoptosis of myonuclei and immune cells, the sarcolemma of TUNEL stained sections was, after the first imaging session, re-stained with spectrin. The coverslips were removed and immunostaining was carried out according to manufacturer's protocol, omitting the neutralization of endogenous peroxidase, as this had been carried out previously. In addition, in 3 cases no adherent granzyme B⁺ or CD8⁺ cells were detected in proximity of fibres with apoptotic myonuclei in the first staining session, and therefore the remaining adjacent sections were stained again with these antibodies.

### Digital imaging and quantification

A digital image, using an AxioCam MRc-digital camera and Zen software (Carl Zeiss Microimaging GmbH, Jena, Germany), was obtained of each section, harbouring the earlier identified partial invasion. In the patients with DM, an area was identified, showing characteristic myopathic findings, such as a perifascicular atrophy (3 cases), other forms of perifascicular pathology (4 cases) or an inflammatory infiltrate (9 cases). A random area was chosen in the control biopsies. In each of the consecutive sections of the biopsies, the corresponding centre point was identified in the image. Using the Zen software, a circle with an area of 1.37 mm$^2$ around this centre point was drawn, and the stained cells and fibres in this area were counted. Spectrin stain was used to demarcate fibres in order to facilitate identification and counting, and the number of stained fibres and inflammatory cells was presented as a ratio per 100 fibres.

### Statistical analyses

Statistical analysis was performed using the Prism program (Graphpad Software). The majority of data were not normally distributed; therefore, the Kruskal-Wallis test was used, followed by Dunn's multiple comparison test. The Pearson normality test, however, confirmed normal distribution of the CD163/ CD68 ratio, hence the parametric ANOVA was used, followed by Tukey's multiple comparisons test. P-values $< 0.05$ were considered significant and a prerequisite for performing the post-test.

### Ethical approval

The biopsies had been taken from patients for diagnostic purposes. Written informed consent was obtained for using remaining tissue for research. The study was performed in accordance with relevant guidelines and regulations, and was approved by the regional Ethical Board in Linköping (ref. no. M58-07).

## Results

### IIM is associated with alteration in the expression of Bcl-2 and of the CD163/CD68 ratio

To provide an overview of apoptosis/anti-apoptosis and macrophage markers in the pathological processes, we first analysed the staining pattern in a circular area (1.37 mm$^2$) surrounding partial invasions, and in areas (of the same size) containing an inflammatory infiltrate or a perifascicular pathology, in patients with DM. The selected areas of the partial invasion cases (pi-cases) contained 93–378 fibres (median 225), and of the DM-patients 132–414 (median 250) fibres. The findings of the IIM groups were also compared with the stains in a random area of the same size (containing 192–252 fibres (median 232)) in control biopsies, from patients without muscle disease.

There was a lower proportion of fibres expressing Bcl-2 in the two groups with inflammatory myopathies compared to controls (Fig 1A). The proportion of fibres expressing FAS, as well as fibres expressing HSP70, was higher in the pi-patients than in the controls, while the difference between DM-patients and the controls was not statistically significant (Fig 1B and 1C). Both the absolute numbers of CD68$^+$ and CD163$^+$ cells, and the ratio of CD163$^+$ and CD68$^+$ cells, were higher in the groups with inflammatory myopathies, compared to controls (Fig 1D–1F). Furthermore, the staining pattern of the CD68$^+$ and the CD163$^+$ cells indicated an expanded cytoplasm compatible with a dendritic morphology, which was not seen in the non-inflammatory controls cases (S1 Fig). In 4 out of 10 pi-cases, 5 necrotic fibres were found, and in 5 out of 10 DM cases, 14 necrotic fibres, whereas no necrotic fibre was detected in the controls.

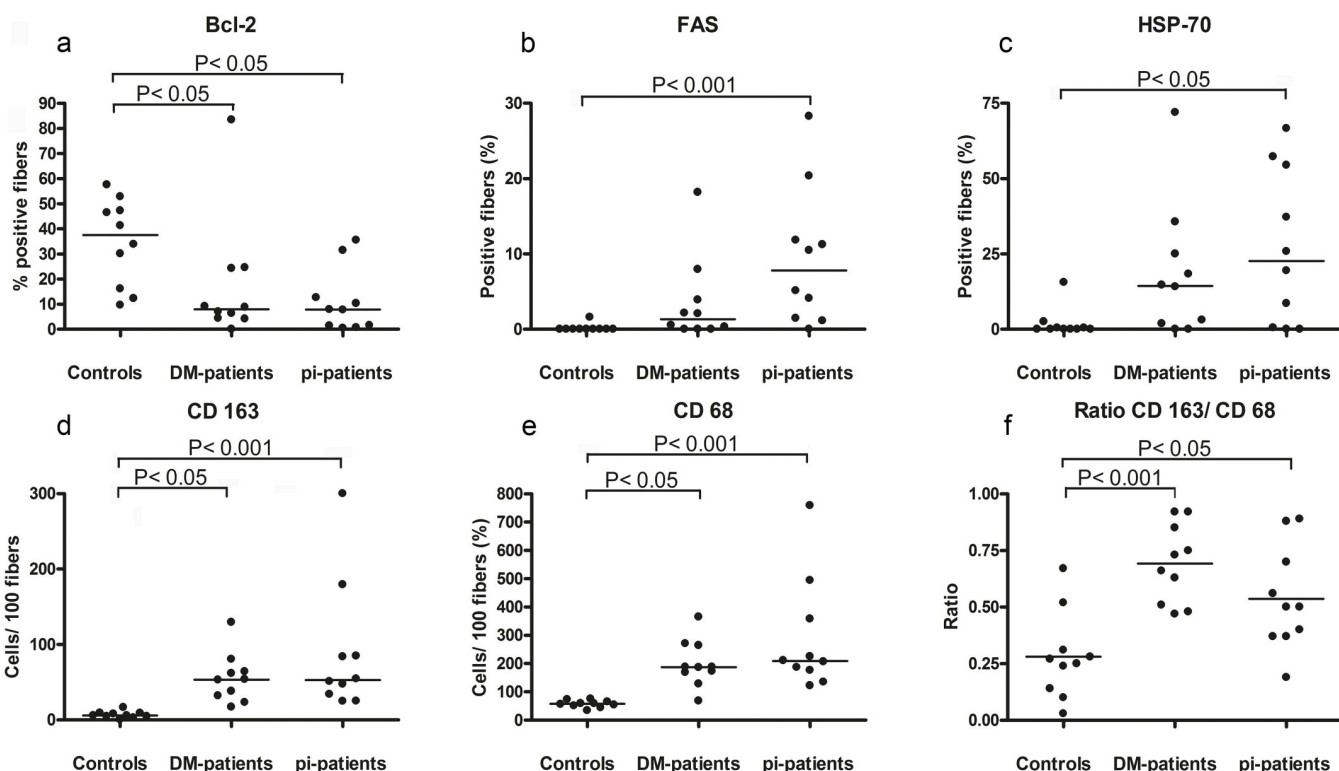

**Fig 1. Frequency of stained fibres and cells.** The graphs show the percentage of Bcl-2 (a), FAS (b) and HSP70 (c) stained muscle fibres, as well as number of CD68$^+$ (d), CD163$^+$ (e) cells per 100 fibres, followed by the CD163$^+$/ CD68$^+$ cell ratio (f) in biopsies from controls (n = 10), dermatomyositis (DM; n = 10) and partial invasion (pi; n = 10).

## Characteristics of partial invasion

In the selected area of the pi-cases (n = 10), 20 partial invasions were detected. The inflammatory cells were in all cases surrounded by a semi-opaque substance, tentative of a common matrix (Fig 2). Most infiltrates (18/20) showed clear signs of invasion also of a neighbouring fibre, and in a majority of the cases (13/20), the part of the infiltrate, directly interacting with the fibre, had a broad base (Fig 2A and 2B), although other morphological appearances of infiltrate-fibre interaction were also noted (Fig 2C and 2D).

The spectrin stain, used in the diagnostic routine to detect membrane damage [47], showed undisrupted expression of the fibre membrane in 14 cases, while the membrane had unstained segments in the remaining 6 cases (Table 2). The merosin stain, which visualizes an integral component of the basal membrane, similarly showed unstained segments, or indicated splitting of the membrane, in 6 cases. Thirteen invaded fibres stained for fast and 17 for slow myosin heavy chain, while 10 stained with both. In addition to CD8$^+$ cells, granzyme B$^+$ cells were found in all invading infiltrates (Table 3). The infiltrates also contained an abundance of CD68$^+$ cells, and at least a few CD163$^+$ cells. The latter were however mainly located outside the fibre. One of the 20 partially invaded fibres was Bcl-2$^+$, while 8 were FAS$^+$ and 9 HSP70$^+$ (Table 3). In order to illustrate some of the different staining patterns of partial invasion, 3 cases are presented in Fig 3.

## Expression of MHC I but never FAS in fibres with apoptotic nuclei

Within the selected areas of two TUNEL stained sections, separated only by the negative control section (6 μm), there were 8 muscle fibres with a TUNEL$^+$ nucleus found (in 5 of the pi-patients) that could be confirmed to be inside the fibre membrane, after re-staining with spectrin (Table 4). This corresponds to a mean of 0.22 TUNEL$^+$ myonuclei per 100 counted fibres (*i.e.* fibre sections). The purpose of the additional stains of the fibre and the surrounding inflammatory cells was to elucidate a possible apoptosis inducing pathway, as well as

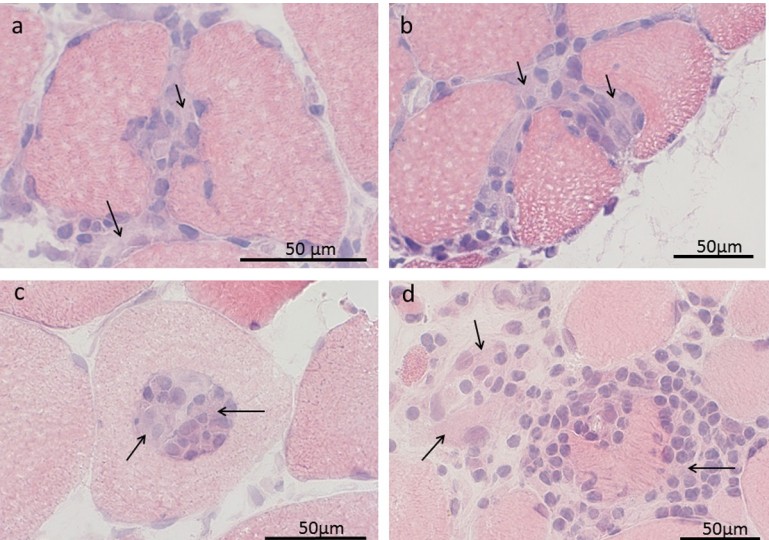

**Fig 2. The histology of partial invasions.** Four cases of partial invasion stained with haematoxylin-eosin (HE) are shown, where the inflammatory infiltrate is surrounded by a semi-opaque material (arrows). In the upper images (a,b) the infiltrate-fibre interaction is broad based and invasion of neighbouring fibres is seen. In the lower left image (c), the infiltrate is (in several consecutive sections) encapsulated within the fibre. The lower right image (d) shows the only case where inflammatory cells indented the fibre at multiple sites.

**Table 3. Characterization of partial invasions (pi).**

| Case | Pi | Bcl-2 | FAS | Hsp70 | Fast | Slow | Spectrin | Merosin | Granzyme B | CD68 | CD163 |
|---|---|---|---|---|---|---|---|---|---|---|---|
| 1 | 1 | – | + | – | + | + | + | – | + | + | + |
|  | 2 | – | – | + | – | + | + | – | + | + | + |
|  | 3 | + | + | – | + | + | + | – | + | + | + |
|  | 4 | – | + | + | + | + | + | – | + | + | + |
| 2 | 5 | – | + | + | + | + | + | – | + | + | + |
|  | 6 | – | – | + | – | + | + | – | + | + | + |
| 3 | 7 | – | – | – | – | + | + | + | + | + | + |
|  | 8 | – | – | – | – | + | – | + | + | + | + |
|  | 9 | – | + | + | + | – | – | + | + | + | + |
| 4 | 10 | – | – | – | + | – | – | + | + | + | + |
| 5 | 11 | – | – | + | + | + | + | + | + | + | + |
|  | 12 | – | – | + | + | + | + | + | + | + | + |
|  | 13 | – | + | + | + | + | + | + | + | + | + |
| 6 | 14 | – | – | + | + | – | – | + | + | + | + |
| 7 | 15 | – | – | – | – | + | – | + | + | + | + |
| 8 | 16 | – | – | – | – | + | + | + | + | + | + |
|  | 17 | – | + | – | + | + | + | + | + | + | + |
|  | 18 | – | – | – | – | + | + | + | + | + | + |
| 9 | 19 | – | – | – | + | + | + | + | + | + | + |
| 10 | 20 | – | – | – | + | + | – | + | + | + | + |

There were 20 partial invasions in the imaged muscle section areas (1.37 mm$^2$) of the 10 cases.

The staining characteristics are shown as positive (+), or negative (-), representing stained or unstained invaded fibres or inflammatory cells. For spectrin and merosin, (+) denotes entire staining and (-) the presence of unstained segments of the membrane.

identifying protective or inducing factors. All 8 fibres had up-regulated MHC I in the sarcolemma and the sarcoplasm. None of these fibres stained for FAS, but one for Bcl-2 and 5 for HSP70. All 8 fibres had CD8$^+$, CD68$^+$ and CD163$^+$ and 7 had granzyme B$^+$ cells, adhering to the sarcolemma (Table 3). In the remaining case, the closest detected granzyme B$^+$ cell was approximately 20μm from the fibre membrane. Notably, 2 fibres showed single CD8$^+$ and granzyme B$^+$ cells inside the cell membrane, close to a TUNEL$^+$ myonucleus. Two examples of fibres with a TUNEL$^+$ myonucleus together with stains of selected marker proteins are shown in Figs 4 and 5. No signs of necrosis, vacuolisation or *classical* partial invasion were found in these fibres. Moreover, spectrin staining showed no unstained section of the cell membranes of these fibres, which all stained negative for MAC. No TUNEL$^+$ fibre nucleus was detected in the selected areas of DM patients or controls. However, TUNEL$^+$ inflammatory cells (S2A and S2B Fig) were detected in both pi- and DM-patients, with a mean of 0.62 and 0.66 per 100 fibres, respectively, but not in the controls.

## Apoptotic nuclei are found almost exclusively in IIM with partial invasion

In order to provide a more reliable basis for the quantification of TUNEL$^+$ fibre nuclei, the entire area of the two TUNEL-stained sections of all patients were studied by light microscopy. The number of fibres in the sections ranged from 954 to 3669 in the pi-patients, from 1019 to 3910 in the DM-patients and from 864 to 2968 in the controls. Forty-five TUNEL$^+$ nuclei were unevenly distributed among all 10 pi-patients (range 1–13), compared to 4 in 3 of the DM-patients and none in the controls. These TUNEL$^+$ nuclei were judged as being fibre nuclei

## Case 1

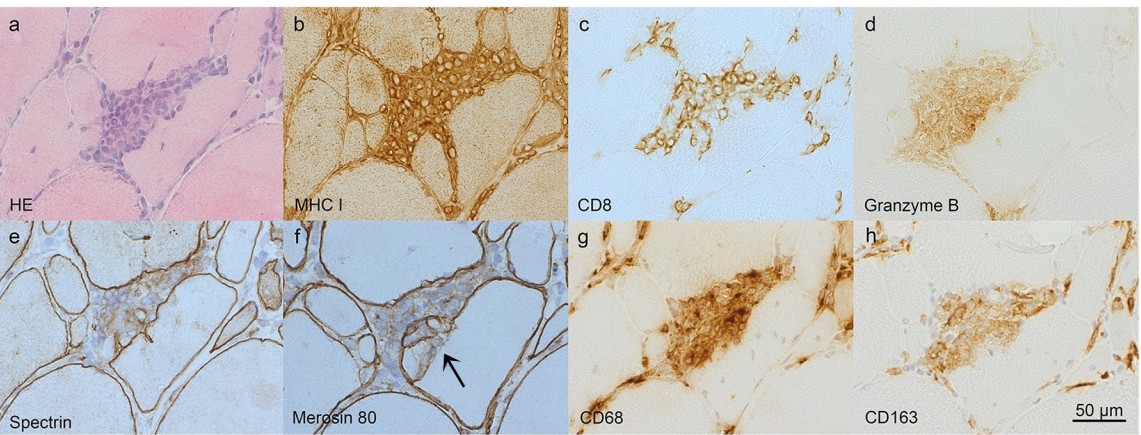

## Case 2

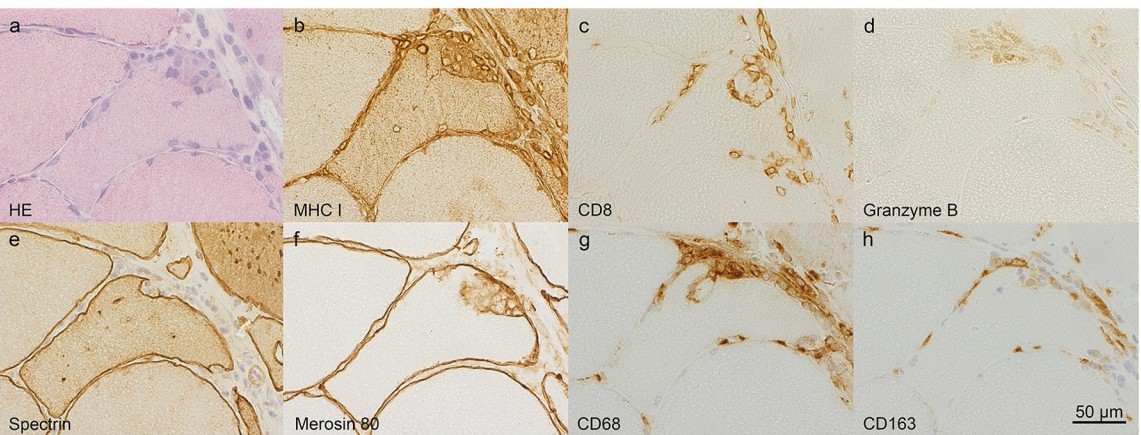

## Case 3

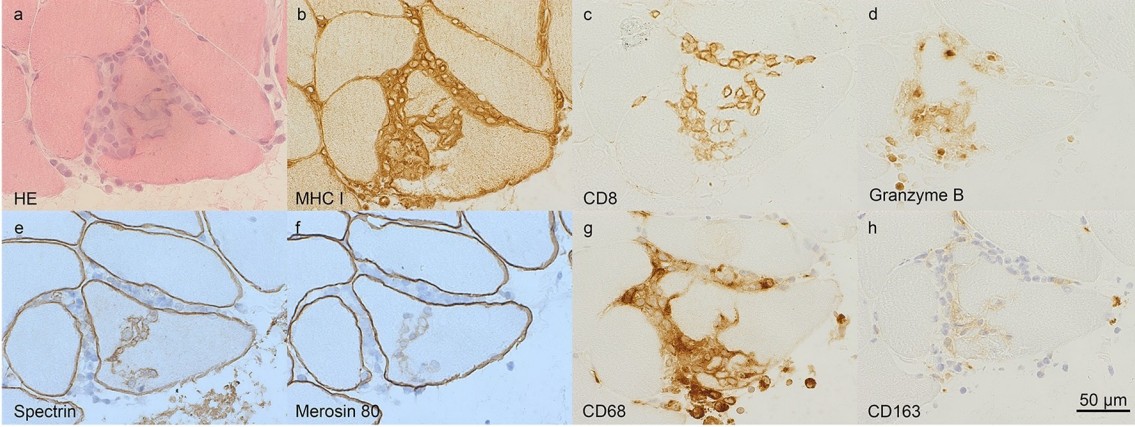

**Fig 3. The immunohistology of partial invasions.** Three cases (1–3) of partial invasions are shown (stained with haematoxylin-eosin (HE), MHC I, CD8, granzyme B, spectrin, merosin 80, CD68 and CD163. Case 1 and 2 have broad based partial invasions in the HE stained sections (case 1-2a), while case 3 has a branching appearance of invading cells (Case 3a). The invaded fibres show sarcoplastic staining of MHC I (case 1-3b). Several invading cells are CD8+ (case 1-3c) and the infiltrate contains granzyme B+ cells (case 1-3d). The

cell membrane is pushed forward into the fibre by the invading cells, as seen by the spectrin stain (case 1-3e), but cases with discontinuity of cell membrane staining are also found (exemplified by case 3e, arrow). The inflammatory cells traverse the basal membrane (case 1f, arrow), which often seems to retain the original form of the fibre (case 2-3f), but is in some cases split (1-2f). There are CD68[+] cells (case 1-3g) among the invading cells. CD163[+] cells are seen in the infiltrate, but mainly outside the fibre (case 1-3h).

based by their localisation and morphology, but were not confirmed by spectrin re-staining of the fibre membrane (S2 Fig), as were the TUNEL[+] myonuclei in the selected areas. In the DM-cases, 2 of the 4 TUNEL[+] fibres were found in a section which also contained a (*sic*) partial invasion. In 3 pi-cases a fibre with two apoptotic nuclei was found in the same section (exemplified in S2D Fig). No TUNEL positivity was found in the necrotic fibres of the selected areas. However, when studying the complete sections, rare necrotic fibres showed different grades of diffuse sarcoplasm staining (S3 Fig), whereas others only showed TUNEL[+] inflammatory cells invading the fibre (S3 Fig).

## Discussion

The presence of apoptosis in IIM has long been debated and its pathogenic role in these diseases considered unlikely [22]. Using the TUNEL assay we detected apoptotic muscle nuclei in inflammatory myopathies, which is in line with earlier studies [20, 21]. Other studies, using similar methods, did not detect apoptosis in IIM (or an increase, compared to controls) [19, 30, 48–50]. These studies, however, showed important differences, compared to our study, regarding presence of partial invasions, size of investigated area and main focus of the study, which may explain this discrepancy.

Interestingly, our results showed that apoptotic nuclei were present almost exclusively in biopsies containing partial invasion, and that the presence of apoptosis was strongly associated with MHC I expressing muscle fibres that did not show the classical signs of partial invasion. However, these fibres did have adherent inflammatory cells expressing CD8[+], granzyme B[+] as well as CD68 and CD163. The two major theories of how granzyme B gain entry into the cytosol of the target cell differ in respect of whether a passive transfer via pores (perforin) is possible or an active transport in necessary [51]. The identification of CD8[+] and granzyme B[+] cells in the sarcoplasm in 2 of 8 cases in this study suggests yet another *in vivo* mechanism of granzyme-target cell entry. Taken together, our findings lend support to an immune-mediated mechanism leading to apoptosis. This process seems to occur in muscle affected by partial invasion, but apparently constituting a parallel process. We would like to point out that double staining of the inflammatory cells was not possible, which means that it cannot be ruled out

**Table 4. Immunohistochemical characterization\* of fibres with apoptotic nuclei (TUNEL[+]) and adherent inflammatory cells.**

| No | MHC1 | MAC | Bcl-2 | Hsp70 | FAS | Fast | Slow | CD8 | Granzyme B | CD68 | CD163 |
|----|------|-----|-------|-------|-----|------|------|-----|------------|------|-------|
| 1 | + | − | − | + | − | + | + | + | + | + | + |
| 2 | + | − | − | + | − | − | + | + | + | + | + |
| 3 | + | − | − | + | − | − | + | +[**] | +[**] | + | + |
| 4 | + | − | + | + | − | − | + | + | + | + | + |
| 5 | + | − | − | − | − | − | + | + | + | + | + |
| 6 | + | − | − | − | − | − | + | + | + | + | + |
| 7 | + | − | − | + | − | − | + | + | − # | + | + |
| 8 | + | − | − | − | − | + | − | +[**] | +[**] | + | + |

\**i e* expression (+) or no expression (−),

\*\*CD8[+] and granzyme B[+] cells located inside the fibre membrane, # nearest detected granzyme B[+] cell located 20 μm from the fibre.

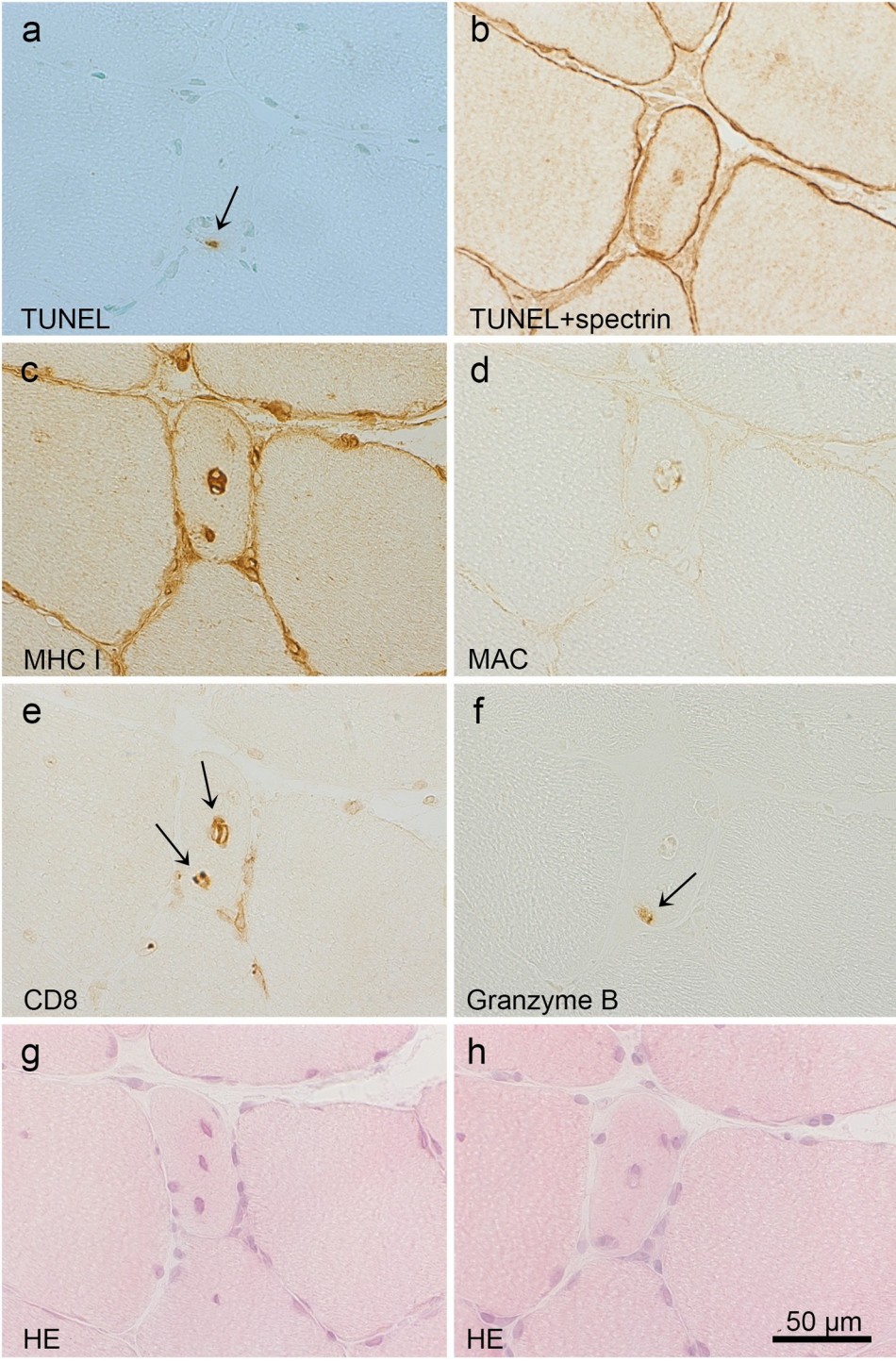

**Fig 4. Fibre with TUNEL$^+$ myonucleus and sarcoplamic CD8$^+$ and granzyme B$^+$ cells.** A TUNEL$^+$ nucleus* is shown (a, arrow), which has a morphology of a myonucleus and is located inside the sarcolemma (b). The sarcoplasm of the fibre stains for MHC I (c), but not MAC (d). CD8$^+$ cells are seen inside the fibre (e, arrows), as is one granzyme B$^+$ cell (f, arrow), which is located close to the TUNEL$^+$ myonucleus. The HE stained sections (g,h) adjacent to the above sections (a-f) do not show signs of classical partial invasion (or necrosis). *Fragmented DNA identifying an apoptotic nucleus stains brown, whereas not affected nuclei stain green (Methyl green).

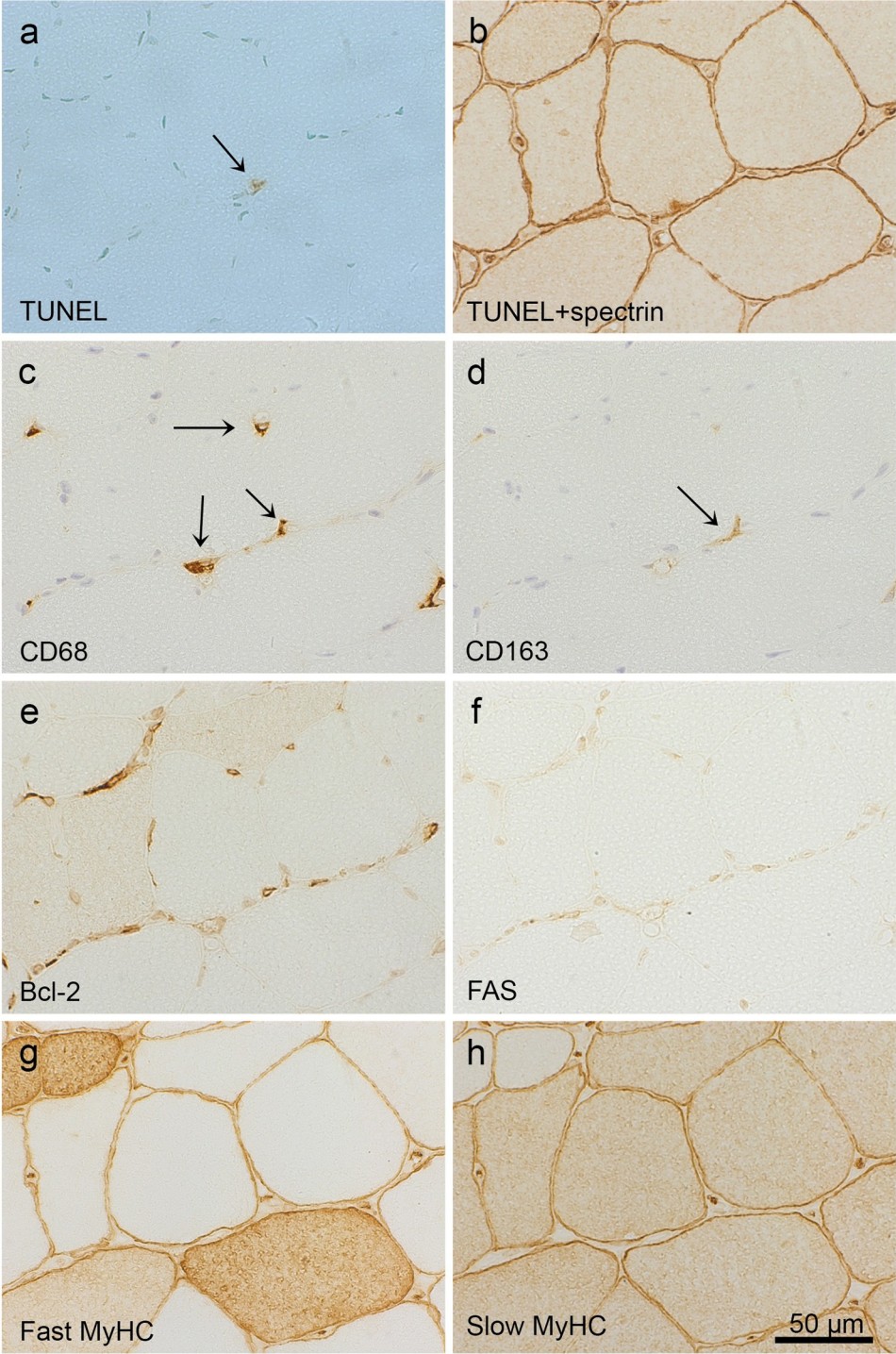

**Fig 5. Fibre with TUNEL+ myonucleus and adherent CD68+ and CD163+ cells.** A fibre with a TUNEL+ nucleus (a, arrow), located inside the cell membrane (b), is shown. The fibre has CD68+ (c) and CD163+ cells (d) adhering to the sarcolemma (arrows). The fibre does not stain for Bcl-2 (e), FAS (f) or fast myosin heavy chain (g), but for slow myosin heavy chain (h).

that NK-cells, although rare in endomysial infiltrate [6], may be represented as some of the granzyme B+ cells. Further, other pathological subgroups of IIM than those with partial

invasion and pathology characteristic of DM were not studied. The presence or absence of apoptosis in the IIM subgroups classified as immune-mediated necrotizing myopathy, unspecific myositis and the recently emerged group characterized by perifascicular pathology, associated with an antisynthetase syndrome [52] should be addressed in further studies.

The absence of FAS in fibres with apoptotic nuclei, which is in agreement with earlier studies [16, 19, 30], together with the presence of granzyme B[+] cells, strongly favours a cytotoxic mode of apoptosis induction rather than a FAS-mediated mechanism. Other proteins secreted by the CD8[+] cytotoxic T cells, such as other granzymes [53] or other molecules [15, 54] may be alternative, albeit less likely, candidates of inducing apoptosis.

Apoptotic nuclei have earlier been demonstrated in muscle from IBM patients using sequential stains [20]. As pointed out by the authors of this report, the presence of single apoptotic nuclei in muscle fibres does not *per se* prove a general apoptotic process in the fibre. In addition, the low number of apoptotic fibre nuclei in IIM has been interpreted as lack of pathogenic relevance [20, 22]. However, the short time window of DNA fragmentation detected by TUNEL-staining, in the range of 1–3 hours, suggests that its expected appearance in normal muscle would be a rare finding [45]. Therefore, our finding of 3 fibres with 2 TUNEL[+] myonuclei in the same section is an event that unlikely would occur by chance only, and indicates that the apoptotic process may be affecting a segment of the fibre.

Our finding (mean of 0.22 TUNEL[+] myonuclei per 100 fibres) is within the range of earlier studies detecting apoptotic myonuclei in IIM, with means calculated from presented data from 0.079 [20] to 0.36 [21]. The differences in frequency may be accounted for by of different methods, and by the areas examined, in our case the area surrounding partial invasion. In line with other studies [8, 50], we did not detect apoptotic myonuclei in partially invaded myofibres. Notably, in the study of Hutchinson et al. [20], investigating PM and IBM, 2 of 100 fibres with TUNEL[+] myonuclei were found in a partial invasion. Constituting such a small fraction, it was however concluded that apoptosis is an unlikely fate of such affected fibres. This conclusion agrees with our findings, indicating that apoptosis may be a parallel form of fibre degeneration, aside from the fibre disintegration seen in partial invasion. The rarity of observed apoptotic myonuclei in IIM, in spite of a repeatedly reported presence of granzyme B and perforin [14, 15], has been attributed to presence of anti-apoptotic proteins in myofibres, such as Bcl-2 [22]. An alternative explanation could be properties of the cell membrane of myofibres, impeding the entry of granzyme B, or granzyme B secreting lymphocytes, which were observed in two TUNEL[+] fibres in this study.

The landmark investigations by Arahata and Engel [6, 55], using immunohistochemistry, showed that the majority of the invading cells of partial invasion are CD8[+] cytotoxic cells, followed in frequency by CD68[+] macrophages. Ultrastructural studies by the same authors [8] showed that the infiltrating cells push the morphologically intact cell membrane into the fibre, and although membrane reparative signs where noted, classical signs of fibre necrosis or apoptotic nuclei were not observed. The immunohistological findings in our study generally agree with this description. However, the detection of sections of cell membrane, lacking spectrin in 6 of 20 cases, is likely to signify a disturbed cell membrane function, facilitating the sarcoplasmic entry of molecules present in the invading infiltrate. Consecutive HE-stained sections of partial invasions showed that the invading inflammatory cell infiltrate, apparently surrounded by a matrix, often also invaded neighbouring fibres, indicating an acquisition of an invasive potential of the infiltrate. Such invasive potential is likely triggered by factors in the microenvironment of the infiltrate [56]. Granzyme B, which also may be produced by myeloid dendritic cells [57], has several substrates in the extracellular matrix [57] and is able to cleave and activate several pro-inflammatory cytokins [58]. Thus, granzyme B may, apart from its pro-apototic effect, also affect the properties, such the invasiveness, of the infiltrate.

The macrophages present in IBM and PM have been reported to express the BDCA-2 antigen and show a branching morphological appearance, typical of myeloid dendritic cells [24]. Here, this morphological appearance was noted in the CD68 and CD163 stains as well. It is unknown if these cells, with macrophage and myeloid dendritic properties, are able to activate T cells locally by presenting antigen [24], and if they secrete directly acting inflammatory molecules [25]. However, these cells are likely important in the muscle specific form of immune attack studied here. In this study, the CD163$^+$ macrophages constituted a small fraction of all macrophages (CD68$^+$) in control muscle, but their relative ratio to CD68$^+$ macrophages was increased in inflammatory myopathies. Further studies, using multiple cell population and activation markers, may help to clarify the immunologic function of these cells, with seemingly overlapping macrophage and myeloid dendritic cell properties. Concomitant immune labelling of cells in muscle and blood from patients may further clarify if CD163$^+$ macrophages are recruited to counteract the inflammatory reaction [26], or are a result of a type switch of present M1 macrophages in muscle during the disease process.

Bcl-2 expression has been shown to protect against different kinds of cell death [59–62], including pathways mediated by granzymes [34]. The present findings of a constitutive expression of Bcl-2 in healthy muscle, and a lower expression of Bcl-2 in IIM, confirm results from our previous study [18] and the study by Ibi et al. [63]. The Bcl-2 expression of the pi-cases in the present study was similar to the expression found in polymyositis patients in our previous study [18], with a median of 7.8 and 5.0%, positive fibres, respectively [18]. Notably, in the present study, the controls (median age 46 yrs.) showed clearly lower expression compared to the healthy volunteers (median age 28 yrs.), in the previous study, with Bcl-2 median values of 38% and 91%, respectively. This agrees with a reported lower expression of Bcl-2 in healthy individuals with increasing age [64], which may affect the vulnerability of muscle towards IIM and other myodegenerative processes. We observed one fibre with a TUNEL$^+$ nucleus, which also expressed Bcl-2, indicating that its expression does not preclude an apoptotic process in muscle fibres. HSP70 expression was absent or faint in most control patients, but higher in the majority of patients with inflammatory myopathies, particularly in pi-patients. However, the great variation within the groups makes interpretation difficult, other than that, HSP70 is unlikely to have a specific role in the two studied types of fibre degeneration.

In conclusion: Apoptosis of muscle fibre nuclei is present in inflammatory myopathies with partial invasions, and these processes occur in the same area, but rarely in the same fibre. The affected fibres express MHC I and are surrounded by CD8$^+$ T-cells, macrophages and granzyme B$^+$ cells. The findings collectively support that apoptosis, induced by cytotoxic CD8$^+$ T-cells in IIM, may be a mechanism activated in parallel with the fibre disintegration seen in partial invasion. In addition to its apoptosis inducing potential, granzyme B may have other important roles in IIM with partial invasion. Inflammatory cells with macrophage/myeloic dendritic properties are also present in infiltrates invading muscle fibres and adhere to fibres with apoptotic myonuclei.

## Supporting information

**S1 Dataset.**
(XLS)

**S1 Fig. The morphology of macrophages in IIM.** The stains of the CD68$^+$ and the CD163$^+$ cells in the cases with partial invasion (a) and dermatomyositis (b) indicate an expanded cytoplasm, compatible with a dendritic morphology, which is not seen in the non-inflammatory control cases (c and d [close-up]).
(TIF)

**S2 Fig. TUNEL-stain of inflammatory cells and myoneclei.** Two examples of TUNEL⁺ inflammatory cells (arrows) are shown, one close to a partial invasion with a branching appearance (a) and one in a case with dermatomyositis (b). When investigating the whole sections of the cases with partial invasion 45 TUNEL⁺ myonuclei were found, as judged by their morphological appearance and location, as exemplified here (c). One of the 3 cases in which 2 TUNEL⁺ myonuclei in the same fibre where found is shown (d).
(TIF)

**S3 Fig. TUNEL-stain of necrotic fibres.** The vast majority of the necrotic fibres showed no TUNEL-stain (a). But rare necrotic fibres did show varying degrees of diffuse sarcoplasmic stain (b,c). The nuclei of some invading inflammatory cells stained TUNEL positive (d).
(TIF)

## Acknowledgments

We thank Erik Edston for supplying us of with heart tissue as a positive control for the TUNEL-method.

## Author Contributions

**Conceptualization:** Olof Danielsson, Bo Häggqvist, Liv Gröntoft, Karin Öllinger, Jan Ernerudh.

**Data curation:** Olof Danielsson, Bo Häggqvist, Liv Gröntoft.

**Formal analysis:** Olof Danielsson, Bo Häggqvist, Karin Öllinger, Jan Ernerudh.

**Funding acquisition:** Olof Danielsson.

**Investigation:** Olof Danielsson, Bo Häggqvist, Liv Gröntoft.

**Methodology:** Olof Danielsson, Bo Häggqvist, Liv Gröntoft.

**Project administration:** Olof Danielsson, Liv Gröntoft.

**Software:** Olof Danielsson, Bo Häggqvist.

**Supervision:** Karin Öllinger, Jan Ernerudh.

**Validation:** Olof Danielsson, Bo Häggqvist, Liv Gröntoft.

**Visualization:** Olof Danielsson, Bo Häggqvist, Liv Gröntoft.

**Writing – original draft:** Olof Danielsson.

**Writing – review & editing:** Olof Danielsson, Bo Häggqvist, Liv Gröntoft, Karin Öllinger, Jan Ernerudh.

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
