## [Decision Letter · Decision Letter 0]

9 Mar 2020

PONE-D-19-35903

Partial invasion and apoptosis in idiopathic inflammatory myopathies; parallel processes mediated by CD8+ cytotoxic T cells

PLOS ONE

Dear Dr Danielsson,

Thank you for submitting your manuscript to PLOS ONE. After careful consideration, we feel that it has merit but does not fully meet PLOS ONE’s publication criteria as it currently stands. Therefore, we invite you to submit a revised version of the manuscript that addresses the points raised during the review process.

Neither reviewer indicated that your results supported the conclusions you drew from them. The first reviewer mentioned that your sample size was too small to draw such robust conclusions and reviewer 2 reported that you should have used non-parametiric methods of statistical validation.

We would appreciate receiving your revised manuscript by Apr 23 2020 11:59PM. To enhance the reproducibility of your results, we recommend that if applicable you deposit your laboratory protocols in protocols.io, where a protocol can be assigned its own identifier (DOI) such that it can be cited independently in the future. For instructions see: http://journals.plos.org/plosone/s/submission-guidelines#loc-laboratory-protocols

We look forward to receiving your revised manuscript.

Kind regards,

Alfred S Lewin, Ph.D.

Academic Editor

PLOS ONE

Journal Requirements:

2. In your Methods section, please provide additional information about the participant recruitment method and the demographic details of your participants. Please ensure you have provided sufficient details to replicate the analyses such as: a) the recruitment date range (month and year), b) a description of any inclusion/exclusion criteria that were applied to participant recruitment, c) a table of relevant demographic details, d) a statement as to whether your sample can be considered representative of a larger population, and e) a description of how participants were recruited.

Reviewers' comments:

Reviewer's Responses to Questions

**Comments to the Author**

1. Is the manuscript technically sound, and do the data support the conclusions?

Reviewer #1: Partly

Reviewer #2: Partly

2. Has the statistical analysis been performed appropriately and rigorously? 

Reviewer #1: Yes

Reviewer #2: No

3. Have the authors made all data underlying the findings in their manuscript fully available?

Reviewer #1: Yes

Reviewer #2: Yes

4. Is the manuscript presented in an intelligible fashion and written in standard English?

Reviewer #1: Yes

Reviewer #2: No

5. Review Comments to the Author

Reviewer #1: Olof Danielsson et al, analysed muscle biopsies from 10 IIM patients, 10 DM patients and 10 donors without muscle disease. They performed TUNEL assays and immunohistochemistry on muscle biopsies. The authors have four main conclusions :

1. IIM is associated with alteration in the expression of Bcl-2, FAS, HSP70, CD163 and CD68.

2. Apoptotic nuclei express MHC I but not FAS

3. Apoptotic nuclei are found almost exclusively in IIM with partial invasion

4. Majority of invading cells in partial invasions are CD8+ cytotoxic cells

Authors suggest that partial invasion and apoptosis are likely mediated by cytotoxic CD8 cells

Major comments :

1. Title of the manuscript is « Partial invasion and apoptosis in idiopathic inflammatory myopathies; parallel processes mediated by CD8+ cytotoxic T cells », nevertheless the authors have only 10 cases of IIM, including 3 IBM def (1 poss), 1 overlap def, PM….. Following the 119th ENMC classification, IIM also include IMNM and DM. Why is DM placed separately in a different group, why is there no IMNM case. Sample size is a little small to conclude.

2. Line 299 : « In the DM-cases, 2 of the 4 TUNEL+ fibres 300 were found in a section which also contained a (sic) partial invasion. ». How many DM patients have partial invasions.

3. Line 190 : « IIM is associated with alteration in the expression of Bcl-2, FAS, HSP70 and of CD163/CD68 ratio ». Authors could consider changing this sentence, as no significant differences in the expressions of FAS and HSP70 are observed between DM and controls (DM are IIM).

4. Table 1 : Authors show necrotic fibres in 4 out of 10 patients. Surprisingly necrotic fibres are not TUNEL+ fibres and vice versa. It has been described that TUNNEL assay does not discrimate apoptosis from necrosis (BETTINA GRASL-KRAU et al, HEPATOLOGY, May 1995). How can the authors explain these findings.

5. The lack of expression of FAS in apoptotic nuclei is indeed surprising. Nevertheless, considering the relatively low mean frequency of TUNEL+ myonuclei ( 0.22 per 100 fibres) and the presence of TUNEL+ fibres in only 5 out of 10 patients, it is possible that these TUNEL+ myonuclei do not have any pathological significance. A greater sample size could help make the results more robust.

6. Authors show increase frequency of FAS+ fibers in pi-patients compared to controls, nevetheless apoptotic fibers do not express FAS. How can the autors explain this, do non-apoptotic fibers express FAS.

7. Line 99 : control samples are young compared to pi-patients, mark difference in age distribution.

Minor comments :

1. Table 1 : Patient 7, change « IMB » to « IBM »

2. How can the authors confirm that partial invasion and apoptosis are mediated by CD8 cytotoxic cells. Can the authors show co-immunostaining of CD8 and granzyme B.

3. Line 231 : « In addition to CD8+ cells, granzyme B+ cells were found in all invading infiltrates (Table 2). » Table 2 shows Antibodies and immunohistochemical procedure, maybe the authors wanted to state Table 3 ?

Reviewer #2: Authors performed immunohistochemical examinations of muscular tissues which were treated with some biological markers including phenotypical, cytotoxic, or enzymatic protein, and apoptosis assay; in addition, they were statistically analyzed between idiopathic inflammatory myopathies (IIM) including polymyositis (PM) or inclusion body myositis (IBM) and controls including dermatomyositis and others. This manuscript has some issues to describe as follows.

１. Turkey’s multiple comparison test is usually performed in the parametric population. In this study, is it appropriate to use Turkey’s test? Otherwise, extracted data should be re-analyzed by the non-parametric statistics.

２. In the figure 1, CD163/CD68 ratio was shown. Can you explain what the frequencies of the double positive cells with CD68 and CD163 were expressed?

３. Authors should indicate the clinical findings related to the musculoskeletal impairment in all participants.

４. It is important to clarify the implication of the pathological characteristics in IIM through showing the correlation between clinical findings and analyzed data based on the basic research. Please demonstrate how the immunopathological findings shown in this study are implicated in the development of clinical impairment. Namely, authors should indicate the relationship between laboratory data/muscular scores and the immunopathological data which were detected in this study. Laboratory data may include serum creatine kinase and/or aldolase, and muscular scores may be manual muscle test (MMT). MMT8 is useful for scoring muscle weakness as the prevalent evaluation of IIM.

５. What kinds of autoantibodies were related to the patients with IIM in this study? Recently, IIM has been recognized as the overall category; moreover, each patient should be categorized into each classification of inflammatory myositis according to the disease-specific antibody. If possible, it is ideal to show the relationship between specific antibody and analyzed data.

６. Immuno-mediated necrotizing myopathy (IMNM) should be differentiated from other types of PM or IBM. Author should demonstrate the differences of characteristics between IMNM, other types of PM, and IBM, if applicable.

6. PLOS authors have the option to publish the peer review history of their article (what does this mean?). If published, this will include your full peer review and any attached files.

Reviewer #1: No

Reviewer #2: No

---

## [Author Response · Author response to Decision Letter 0]

23 Jun 2020

The responses are uploaded as a separate file.

Alfred S Lewin, Ph.D.

Academic Editor

PLOS ONE

Journal Requirements:

 We have done our best to comply with the style requirements, but will certainly make any necessary additional corrections. 

2. In your Methods section, please provide additional information about the participant recruitment method and the demographic details of your participants. Please ensure you have provided sufficient details to replicate the analyses such as: a) the recruitment date range (month and year), b) a description of any inclusion/exclusion criteria that were applied to participant recruitment, c) a table of relevant demographic details, d) a statement as to whether your sample can be considered representative of a larger population, and e) a description of how participants were recruited.

We rewrote the first two paragraphs in the methods section, and added a third, with more detailed information about the recruitment process and the representativeness of the included cases (line 97-141, below), and we added demographic details of all participants in the S1 Table (please see uploaded file). 

Methods

Patients and biopsies

“All biopsies were taken from the anterior tibial muscle for diagnostic purposes. We included stored tissue samples, obtained 1997-2002, with confirmed partial invasion from 10 patients (median age 62 years, range 32-75) with inflammatory myopathies (Table 1), which in a previous study [4] had been classified according to the Amato/European neuromuscular centre (ENMC) classification [7]. These samples were compared with two control groups, comprising 10 patients in each group. As an IIM control group, biopsies from patients with DM were selected; the reason for this choice was that this disease has another type of, more or less pathognomonic, pathological findings [7]. The DM cases (median age 52 years, range17-78) were consecutive biopsies investigated in our lab, where the available clinical and laboratory data allowed the classification of at least probable DM, according to the Amato/ENMC. Eight cases had a definite and two a probable DM according to this classification, and all cases had a definite DM according to the Bohan and Peter. The other control group comprised consecutive cases (median age 46 years, range 39-59), where neuromuscular disease had been excluded, based on biopsy and clinical examination. The cases in these latter two groups were included during April 2015. For demographic details of all cases, see S1 Table.

Sectioning method and selection of cases with partial invasion

In the earlier study [4], muscle biopsies from 36 patients were shown to contain at least one partial invasion. In order to detect and verify strict biopsy criteria stated by Amato/ENMC [7], 35 consecutive cryosections had been made from each biopsy, and every 7th section (6 µm) stained with haematoxylin eosin (HE). Partial invasions were first identified by the morphologic appearance in the HE stained sections and then confirmed by the following three sections, stained with antibodies against MHC I, CD8 and membrane attack complex (MAC). MAC was included to add sensitivity to detect myofibre necrosis [43]. For the present study, the frozen consecutive sections (-70º C) of the biopsies were thawed, and the HE stained sections were again studied in a Zeiss AXIO Imager light microscope. From the 36 cases, we selected those where the partial invasion were present in at least 3 HE sections (n = 10), and used the intermediate sections for immunohistochemistry and the TUNEL-assay. 

Reflections on the representativeness of the included cases

The requirement that the partial invasion had to be detectable in 3 HE stained sections, had the effect that partial invasions with a smaller size than approximately 150 µm were not included. We have not found any report voicing the possibility that partial invasions of different sizes may differ in a qualitative manner, but it cannot be excluded. We earlier showed that partial invasions is present in many types of IIM [4]. However, it is more common and more frequent in IBM, followed by PM [44]. The distribution of diagnoses of investigated pi-cases is comparable with that found in the 36 cases in our 6 year IIM cohort [4] (Table 1). With the above reservation of pi-size, the results are thus considered representative for IIM with pi, not of any defined disease subgroup.”

Reviewer 1

Reviewer #1: Olof Danielsson et al, analysed muscle biopsies from 10 IIM patients, 10 DM patients and 10 donors without muscle disease. They performed TUNEL assays and immunohistochemistry on muscle biopsies. The authors have four main conclusions :

1. IIM is associated with alteration in the expression of Bcl-2, FAS, HSP70, CD163 and CD68.

2. Apoptotic nuclei express MHC I but not FAS.

3. Apoptotic nuclei are found almost exclusively in IIM with partial invasion

4. Majority of invading cells in partial invasions are CD8+ cytotoxic cells

Authors suggest that partial invasion and apoptosis are likely mediated by cytotoxic CD8 cells

Major comments :

1. Title of the manuscript is « Partial invasion and apoptosis in idiopathic inflammatory myopathies; parallel processes mediated by CD8+ cytotoxic T cells », nevertheless the authors have only 10 cases of IIM, including 3 IBM def (1 poss), 1 overlap def, PM….. Following the 119th ENMC classification, IIM also include IMNM and DM. Why is DM placed separately in a different group, why is there no IMNM case. Sample size is a little small to conclude.

The presence of partial invasion (pi), according to strict criteria, was our method to identify IIM-cases, with an immune response characterized by CD8+ cytotoxic T cells. The reason for choosing dermatomyositis, for comparison, was that in this disease (or group of diseases) the immunological response is different, and only rarely includes partial invasion. Immune-mediated necrotizing myopathy is pathologically dominated by necrosis, and, in the majority of cases, the complete absence of inflammatory infiltrates. We were mainly interested in apoptosis-inducing mechanisms induced by inflammatory cells, and although plausible hypotheses have been presented, the immunopathology of IMNM is at this point less clear (1, 2), thus we decided to not include IMNM. We have now added a comment of IMNM in the introduction section (line 55-61): 

“Another widely accepted pathology based subgroup of IIM is immune-mediated necrotizing myopathy (IMNM), also called necrotizing autoimmune myopathy (NAM) (3) [in text ref. 7]. There is an ongoing discussion about further myopathological subdivisions of these diseases (4, 5) [in text 5, 9]. However, partial invasion is a feature that is common to both IBM and PM (strictly defined), and extensive sectioning of muscle tissue shows that partial invasion is sometimes encountered also in overlap syndromes, and, rarely, in classical DM (6, 7) [in text 4, 6]. This raises the question whether there are further shared pathogenic processes associated with the feature of partial invasion.”

As a response of the editor’s requirement of additional information about the participant recruitment method and the demographic details of your participants, we rewrote the first two paragraphs in the methods section, and added a third. We included a comment explaining our choice of IIM control group in the method section (line 101-103). 

 “As an IIM control group, biopsies from patients with DM were selected; the reason for this choice was that this disease has another type of, more or less pathognomonic, pathological findings (3) [in text 7].

We think that there is much evidence supporting the view that there are more than one pathologic process present, not just in IBM, but also in the other IIM. Also, several non-immune mechanisms have been suggested (8, 9). The studies of Arahata and Engel (7, 10) showed typical immunopathology of the different IIM, but no absolute disease correlation. Our earlier study highlighted this pathological overlap (6), which may be of importance when using pathology in classifications of IIM. In this study our focus was the pathological process with a known potential to induce apoptosis, rather than a classified disease entity.

In response to the editor’s urge, for a statement about the representativeness of our sample) in the method section, we added a paragraph, where we explained this more clearly (line 135-141): 

“The requirement that the partial invasion had to be detectable in 3 HE stained sections, had the effect that partial invasions with a smaller size than approximately 150 µm were not included. We have not found any report voicing the possibility that partial invasions of different sizes may differ in a qualitative manner, but it cannot be excluded. We earlier showed that partial invasions is present in many types of IIM (6) [in text 4]. However, it is more common and more frequent in IBM, followed by PM (11) [in text 44]. The distribution of diagnoses of investigated pi-cases is comparable with that found in the 36 cases in our 6 year IIM cohort (6) [in text 4] (Table 1). With the above reservation of pi-size, the results are considered representative for IIM with pi, while not of any defined disease subgroup.”

Regarding the number of cases: The discussion about the significance of partial invasion in diagnosing IIM, and the controversy of PM as a diagnostic entity (12-14), emphasized the need of using strict criteria for partial invasion. This resulted in a modest number of cases, however similar to earlier studies. Although the detected TUNEL+ myonuclei in the circled areas of pi-patients clearly contrasted to the absence in DM-cases, we felt, as pointed out by the reviewer, that more data would be desirable. That was the reason why the whole sections were studied for TUNEL+ myonuclei, (line 303-306): 

“In order to provide a more reliable basis for the quantification of TUNEL+ fibre nuclei, the entire area of the two TUNEL-stained sections of all patients were studied by light microscopy. The number of fibres in the sections ranged from 954 to 3669 in the pi-patients, from 1019 to 3910 in the DM-patients and from 864 to 2968 in the controls.”

 As the method of detecting positive myonuclei differed, in the way that only those myonuclei with typical morphology and a location clearly inside the sarkolemma were accepted (S2 Fig. c,d) in this extended search, we chose to present these results in a separate paragraph.

We think that the similar frequency of detected apoptosis as in earlier studies, the almost complete absence of TUNEL+ myonuclei in DM and the colocalization with granzyme+ cells, lend support to our conclusion. 

2. Line 299 : « In the DM-cases, 2 of the 4 TUNEL+ fibres 300 were found in a section which also contained a (sic) partial invasion. ». How many DM patients have partial invasions.

In these 10 DM cases, this was the only one. Athough considered rare, it was already described by Arahata and Engel (7), and earlier noted by us (6). 

3. Line 190 : « IIM is associated with alteration in the expression of Bcl-2, FAS, HSP70 and of CD163/CD68 ratio ». Authors could consider changing this sentence, as no significant differences in the expressions of FAS and HSP70 are observed between DM and controls (DM are IIM).

We agree. The sentence has been changed (line 203-204): “IIM is associated with alteration in the expression of Bcl-2 and of CD163/CD68 ratio”.

4. Table 1 : Authors show necrotic fibres in 4 out of 10 patients. Surprisingly necrotic fibres are not TUNEL+ fibres and vice versa. It has been described that TUNNEL assay does not discrimate apoptosis from necrosis (BETTINA GRASL-KRAU et al, HEPATOLOGY, May 1995). How can the authors explain these findings. 

We are grateful for this remark. We missed to attach the third supplementary figure file (S3 Fig), now added, which is relevant for this question (see below).

As a consequence of the findings of Grassel-Kraupp et al., Edston et al. (15), who generously supplied us with tissue for positive control, investigated decomposing heart tissue with two different TUNEL-kits. They found that false TUNEL positivity can occur in autolytic fibres with the Cardiotacs kit (due to inadequate fixation), but not with Apotag kit (which we used).

The referred studies, investigating apoptosis in muscle, do not report TUNEL-positivity of necrotic fbres. Olivé et al. specifically state that no necrotic myofibres stained by DNA fragmentation in situ labeling of muscle tissue (16). However, in this study we show that rare necrotic fibres do show TUNEL-stain diffusely in the sarcoplasm (S3 Fig). 

Typical morphologic findings of fibre necrosis are detectable also in the TUNEL-stain of TUNEL- negative fibres, and the stains of the adjacent sections are well suited to detect necrotic fibres (especially the HE and spectrin stains). The reason for including MAC (C5-C9), was to detect supple signs of necrosis. Engel and Biesecker concluded that complement activation is an “invariant concomitant of muscle fibre necrosis” (17). To draw attention to this we have added in the methods section (line 128): “MAC was included to add sensitivity to detect myofibre necrosis (17).” 

5. The lack of expression of FAS in apoptotic nuclei is indeed surprising. Nevertheless, considering the relatively low mean frequency of TUNEL+ myonuclei (0.22 per 100 fibres) and the presence of TUNEL+ fibres in only 5 out of 10 patients, it is possible that these TUNEL+ myonuclei do not have any pathological significance. A greater sample size could help make the results more robust.

For the comment on FAS, please see comment 6.

For the comment on robustness, please see comment 1.

The circled and photographed areas had been chosen because they contained the pathogenic process of interest and allowed these fibres to be studied in all, differently stained, sections. This allowed the frequency of the different pathological findings to be compared. In this area, there were 20 partial invasions, 5 necrotic fibres and 8 TUNEL+ myofibres and 21 TUNEL+ inflammatory cells. When comparing the relative frequency of these finding, one has to consider: 

1. The apoptotic myonuclei were studied in 2 sections. 

2. The TUNEL+ myonuclei are somewhat larger and inflammatory cells are somewhat smaller than the size of one section, and are as a rule, not seen in neighbouring sections.

3. Necrotic fibres and partial invasions are of far greater size, and seen in many sections. A requirement for partial invasionen in this study was that it could be detected in 21 consecutive sections.

4. The time interval, during which a fibre necrosis can be detected is longer than that of a TUNEL+ nucleus (Karpati and Molnar estimated 6-8 hours, until the invasion of phagocytic macrophages (18)), and we estimate that the time process of partial invasion is as least as long. If these assumptions are sound, apoptotic myonuclei are more prevalent than necrosis and partial invasion. This would be reasonable, as myonuclei are plentiful and only serve a limited sarcoplasmic domain.

The question of pathological significance of these signs of muscle fibre demise is relevant, but also applies to other observed pathological changes found in IIM. There is, overall, a dissociation between inflammation and the clinical finding of muscle weakness in IIM (6, 19) and parallel non-inflammatory mechanisms have been identified (20). We think that there are several missing pieces in our understanding of these diseases, and that our findings of myonuclear apoptosis, being present in a pathologically defined subgroup of IIM, but not in the control groups, deserves attention and further study. 

6. Authors show increase frequency of FAS+ fibers in pi-patients compared to controls, nevertheless apoptotic fibers do not express FAS. How can the autors explain this, do non-apoptotic fibers express FAS. 

The expression of FAS in non-apoptotic fibres has been shown in earlier studies (ref. line 335-336), and different explanations have been put forward. Behrens et al. noted a co-expression of FAS and Bcl-2 and suggested a protective role of Bcl-2 (21). In an earlier study (22), we noted that, although both proteins were more commonly expressed on Type 2-fibres, the number of fibres expressing Bcl-2 was inversely related to the number of FAS expressing fibres. This is not evidence against a protective rule on a single fibre level, but it would require other means of apoptosis protection. De Bleecker et al. showed that most FAS+ fibres were NCAM-positive regenerating muscle fibres (23), and mention the function of FAS as a costimulatory factor in the generation of an immune response (24). They considered that a protective effect of Bcl-2 cannot be excluded, but suggested that muscle fibre FAS and Bcl-2 is part of an activated or reactivated regeneration program, that had “little to do with prevention or induction of apoptosis”. We have added this reference in the discussion section (line 335). In the review of Muscle fibre apoptosis in neuromuscular diseases, Dominique Tews also draws attention to the immune-activating role of FAS, and in addition to Bcl-2, mentions the upregulation other anti-apoptotic proteins as a possible explanation for the absence of apoptosis in FAS+ fibres (25).

 We think that our findings with no TUNEL+ myonuclei in FAS+ fibres strengthen the conclusion that FASL-FAS does not induce apoptosis in IIM. One may consider the (perhaps remote) possibility of a time delay of apoptosis in FAS+ fibres, rarely observed because of its short time frame, but we think that our simultaneous observation of a granzyme B+ cell, in the sarcoplasm, next to an apoptotic myonucleus, in the absence of FAS, makes this very unlikely. We think that De Bleecker at al. (23) make a good case for their theory, and agree with the possible explanations put forward by Tews (25).

7. Line 99 : control samples are young compared to pi-patients, mark difference in age distribution.

Age distribution (years):

 pi DM controls

median 63.5 57.3 45.3

SD 12.2 18.9 5.6

mean 61,7 52.3 46.0

As the reviewer points out, the age of the controls is lower than that of the pi-patient. To detect any age influence of the results we performed correlation analysis (Spearman) of the quantitative results within each group. In the control group without muscle disease, most correlation coefficients were close to zero. Only the HPS70 showed a positive correlation of 0.64 (p= 0.047). Here, an outlier with 30% HSP70+ fibres, was the main contributor (one case had 5%, two cases 1% the other six 0%). Had it been excluded, it would have been resulted in r 0.5578 and p= 0.1270. In the DM-group r was -0.12 (p= 0.72) and in the pi-group r was 0.22 (p=0.54).

No significant age correlation was found in the DM-group, where most of the correlations were close to zero.

In the pi-group there were statistically significant positive age correlations for three results: The percentage of FAS+ fibres (r 0.76, p= 0.015), percentage CD68+ cells (r 0.68, p= 0.03) and the percentage of fibres expressing fast myosin heavy chain (r 0.64, p=0.05). Our interpretation is that these correlations do not have any impact on our results. As the expression of these proteins showed very low r values in the other two groups, these correlations may represent type 1-errors.

We have also investigated the presence of TUNEL+ myonuclei in controls in the age groups 20-39 and 60-80 yrs in an ongoing study (not yet published), using the same method and setting (a circled area of the same size). No TUNEL+ myonucleus was found in these control groups. 

Minor comments :

1. Table 1 : Patient 7, change « IMB » to « IBM »

Thank you, now changed.

2. How can the authors confirm that partial invasion and apoptosis are mediated by CD8 cytotoxic cells. Can the authors show co-immunostaining of CD8 and granzyme B.

The finding of granzyme B+ and CD8+ cells in the sarcoplasm, next to a TUNEL+ myonucleus in two cases was an unexpected finding, not described earlier. The cells stained by granzyme B and CD8 are not the same cells. The sections are on 18 and 12 µm, respectively, of either side of the TUNEL/spectrin stained section. It would indeed have been elegant to show a co-stain.

We applied a CD8 antibody, directly conjugated to fluorescent Alexa 488, to earlier stained granzyme B-stained sections, but the previous staining complex prevented binding of the earlier stained cells. Cytotoxic CD8+ and NK-cells are the only known inflammatory cells, known potential to induce apoptosis, mediated by granzyme B. NK-cells are rare in inflammatory infiltrates in IIM (7), but we cannot rule out that an NK-cell also may entered the sarcoplasm, and is the source of granzyme B. We have added a sentence in the discussion section to clarify this (line 331-334):

 “We would like to point out that double staining of the inflammatory cells was not possible, which means that it cannot be ruled out that NK-cells, although rare in endomysial infiltrate (7) [in text 6], may be represented as some of the granzyme B+ cells.”

3. Line 231 : « In addition to CD8+ cells, granzyme B+ cells were found in all invading infiltrates (Table 2). » Table 2 shows Antibodies and immunohistochemical procedure, maybe the authors wanted to state Table 3 ? Yes, thank you. Now changed.

References

1. Preusse C, Goebel HH, Held J, Wengert O, Scheibe F, Irlbacher K, et al. Immune-mediated necrotizing myopathy is characterized by a specific Th1-M1 polarized immune profile. Am J Pathol. 2012;181(6):2161-71.

2. Allenbach Y, Arouche-Delaperche L, Preusse C, Radbruch H, Butler-Browne G, Champtiaux N, et al. Necrosis in anti-SRP(+) and anti-HMGCR(+)myopathies: Role of autoantibodies and complement. Neurology. 2018;90(6):e507-e17.

3. Hoogendijk JE, Amato AA, Lecky BR, Choy EH, Lundberg IE, Rose MR, et al. 119th ENMC international workshop: trial design in adult idiopathic inflammatory myopathies, with the exception of inclusion body myositis, 10-12 October 2003, Naarden, The Netherlands. Neuromuscular disorders : NMD. 2004;14(5):337-45.

4. De Bleecker JL, De Paepe B, Aronica E, de Visser M, Group EMMBS, Amato A, et al. 205th ENMC International Workshop: Pathology diagnosis of idiopathic inflammatory myopathies part II 28-30 March 2014, Naarden, The Netherlands. Neuromuscular disorders : NMD. 2015;25(3):268-72.

5. De Bleecker JL, Lundberg IE, de Visser M, Group EMMBS. 193rd ENMC International workshop Pathology diagnosis of idiopathic inflammatory myopathies 30 November - 2 December 2012, Naarden, The Netherlands. Neuromuscular disorders : NMD. 2013;23(11):945-51.

6. Danielsson O, Lindvall B, Gati I, Ernerudh J. Classification and diagnostic investigation in inflammatory myopathies: a study of 99 patients. The Journal of rheumatology. 2013;40(7):1173-82.

7. Arahata K, Engel AG. Monoclonal antibody analysis of mononuclear cells in myopathies. I: Quantitation of subsets according to diagnosis and sites of accumulation and demonstration and counts of muscle fibers invaded by T cells. Annals of neurology. 1984;16(2):193-208.

8. Henriques-Pons A, Nagaraju K. Nonimmune mechanisms of muscle damage in myositis: role of the endoplasmic reticulum stress response and autophagy in the disease pathogenesis. Curr Opin Rheumatol. 2009;21(6):581-7.

9. Manole E, Bastian AE, Butoianu N, Goebel HH. Myositis non-inflammatory mechanisms: An up-dated review. J Immunoassay Immunochem. 2017;38(2):115-26.

10. Engel AG, Arahata K. Monoclonal antibody analysis of mononuclear cells in myopathies. II: Phenotypes of autoinvasive cells in polymyositis and inclusion body myositis. Annals of neurology. 1984;16(2):209-15.

11. Arahata K, Engel AG. Monoclonal antibody analysis of mononuclear cells in myopathies. IV: Cell-mediated cytotoxicity and muscle fiber necrosis. Annals of neurology. 1988;23(2):168-73.

12. Amato AA, Griggs RC. Unicorns, dragons, polymyositis, and other mythological beasts. Neurology. 2003;61(3):288-9.

13. Chahin N, Engel AG. Correlation of muscle biopsy, clinical course, and outcome in PM and sporadic IBM. Neurology. 2008;70(6):418-24.

14. van der Meulen MF, Bronner IM, Hoogendijk JE, Burger H, van Venrooij WJ, Voskuyl AE, et al. Polymyositis: an overdiagnosed entity. Neurology. 2003;61(3):316-21.

15. Edston E, Grontoft L, Johnsson J. TUNEL: a useful screening method in sudden cardiac death. International journal of legal medicine. 2002;116(1):22-6.

16. Olive M, Martinez-Matos JA, Montero J, Ferrer I. Apoptosis is not the mechanism of cell death of muscle fibers in human muscular dystrophies and inflammatory myopathies. Muscle & nerve. 1997;20(10):1328-30.

17. Engel AG, Biesecker G. Complement activation in muscle fiber necrosis: demonstration of the membrane attack complex of complement in necrotic fibers. Annals of neurology. 1982;12(3):289-96.

18. Karpati G, Molnar, M. J. Muscle fibre regeneration in human skeletal muscle diseases. In: Schaffiano S, Partridge, T., editor. Skeletal Muscle Repair and Regeneration A A Dordrecht, The Netherlands: Sprnger; 2008. p. 200.

19. Dorph C, Englund P, Nennesmo I, Lundberg IE. Signs of inflammation in both symptomatic and asymptomatic muscles from patients with polymyositis and dermatomyositis. Annals of the rheumatic diseases. 2006;65(12):1565-71.

20. Coley W, Rayavarapu S, Pandey GS, Sabina RL, Van der Meulen JH, Ampong B, et al. The molecular basis of skeletal muscle weakness in a mouse model of inflammatory myopathy. Arthritis and rheumatism. 2012;64(11):3750-9.

21. Behrens L, Bender A, Johnson MA, Hohlfeld R. Cytotoxic mechanisms in inflammatory myopathies. Co-expression of Fas and protective Bcl-2 in muscle fibres and inflammatory cells. Brain : a journal of neurology. 1997;120 ( Pt 6):929-38.

22. Danielsson O, Nilsson C, Lindvall B, Ernerudh J. Expression of apoptosis related proteins in normal and diseased muscle: a possible role for Bcl-2 in protection of striated muscle. Neuromuscular disorders : NMD. 2009;19(6):412-7.

23. De Bleecker JL, Meire VI, Van Walleghem IE, Groessens IM, Schroder JM. Immunolocalization of FAS and FAS ligand in inflammatory myopathies. Acta neuropathologica. 2001;101(6):572-8.

24. Lynch DH, Ramsdell F, Alderson MR. Fas and FasL in the homeostatic regulation of immune responses. Immunology today. 1995;16(12):569-74.

25. Tews DS. Muscle-fiber apoptosis in neuromuscular diseases. Muscle & nerve. 2005;32(4):443-58.

Reviewer 2

Reviewer #2: Authors performed immunohistochemical examinations of muscular tissues which were treated with some biological markers including phenotypical, cytotoxic, or enzymatic protein, and apoptosis assay; in addition, they were statistically analyzed between idiopathic inflammatory myopathies (IIM) including polymyositis (PM) or inclusion body myositis (IBM) and controls including dermatomyositis and others. This manuscript has some issues to describe as follows.

１. Turkey’s multiple comparison test is usually performed in the parametric population. In this study, is it appropriate to use Turkey’s test? Otherwise, extracted data should be re-analyzed by the non-parametric statistics. 

We agree that non-parametric statistics are often a good choice for this type of data. Accordingly, for the results shown in the Figures 1a-e the non-parametric Kruskal-Wallis test was used, followed by Dunn´s multiple comparison test. However, the parametric ANOVA, followed by the Tukey´s test was chosen for the ratio CD163+/CD68+, as the deviation of the data from the mean (see below), conformed to a normal distribution, which was confirmed by the Pearson normality test. Ratios, as well as differences (delta values), may often be normally distributed even if the individual values are not. The statistical approach followed the advice from a statistician. 

２. In the figure 1, CD163/CD68 ratio was shown. Can you explain what the frequencies of the double positive cells with CD68 and CD163 were expressed? 

Since no double staining was done, we cannot say for sure that it is the same cell that expresses CD68 and CD163. However, CD68 is an established macrophage pan-macrophage marker, while CD163 is marker of a macrophage subpopulation. We therefore reported the ratio between the subpopulation and the “mother” population. The two populations were stained in adjacent sections (6 µm thick), thus we believe that our data are representative of the relative occurrence of type 2 macrophages.

 ３. Authors should indicate the clinical findings related to the musculoskeletal impairment in all participants. 

We agree that it is relevant to clarify to the reader why two of the patients were classified as not IIM according to Amato/ENMC, but according to Bohan and Peter. In the legends to Table 1 (line 115-116) we added: 

“# These two patients did not have a detectable muscle weakness and could thus not be classified according to Amato/ENMC.” 

We think that the sample size in this study is too small, and the selection of samples not designed to draw any conclusion related to musculoskeletal impairment. In our earlier study (1), we showed that partial invasion (strictly defined) was not exclusively found in IBM and PM, but also in other IIM disease groups, and also found in patients with muscle pathology consistent with IIM, but without muscle weakness (e. g. overlap syndromes). Contrary to the traditional Bohan and Peter classification, the Amato/ENMC requires muscle weakness for IIM classification (except for amyopathic dermatomyositis) (2-4). 

４. It is important to clarify the implication of the pathological characteristics in IIM through showing the correlation between clinical findings and analyzed data based on the basic research. Please demonstrate how the immunopathological findings shown in this study are implicated in the development of clinical impairment. Namely, authors should indicate the relationship between laboratory data/muscular scores and the immunopathological data which were detected in this study. Laboratory data may include serum creatine kinase and/or aldolase, and muscular scores may be manual muscle test (MMT). MMT8 is useful for scoring muscle weakness as the prevalent evaluation of IIM. 

CK was above normal in all IIM-patients of this study, and was one of the criteria for classification. In the legends to Table 1 (line 115) we added: “* The creatine kinase was elevated in all patients.” We agree that MMT8 is a valuable and recommended outcome measure, as concluded by the 213th ENMC International Workshop (5), but the applied classifications, which were decisive for the selection of our cases, only define the pattern, not the degree of muscle weakness.

For this study, there were 19 muscle biopsies from the 10 pi-patients (1-3 per patient). From each patient the biopsy was chosen, which was best suited for classification. That means that the time from disease onset and the time of the analyzed biopsy varied, and in some cases treatment had been given. So we don´t think that any fair correlations can be done between pathology and clinical findings. The study was not designed for this purpose.

５. What kinds of autoantibodies were related to the patients with IIM in this study? Recently, IIM has been recognized as the overall category; moreover, each patient should be categorized into each classification of inflammatory myositis according to the disease-specific antibody. If possible, it is ideal to show the relationship between specific antibody and analyzed data. 

We agree that the advent of MSA and MAA has added a new and important dimension of IIM-classification. However, the focus of this study was the pathological findings considered more or less specific for IBM and PM. The Amato/ENMC classification, which served as a basis for this and our earlier study, does not require autoantibodies for diagnosis, and we did not obtain these data for inclusion. We also think that a different design of the study would have been necessary, to be able to draw any conclusion from the autoantibody status of the patients. 

６. Immuno-mediated necrotizing myopathy (IMNM) should be differentiated from other types of PM or IBM. Author should demonstrate the differences of characteristics between IMNM, other types of PM, and IBM, if applicable.

There were no patients with IMNM in this study. In IMNM inflammatory infiltrates are scarce and, when found, mostly perivascular. We have not found studies reporting partial invasion in IMNM, and we have so far not found partial invasion, fulfilling strict criteria, in this group (only partial necrosis). We therefore did not include IMNM patients in this study.

We realize that we had been too vague in our description. Therefore we rewrote the aim of the study (line 90-93), to clarify that our main focus was to investigate muscle from IIM patients with partial invasion, and use a pathologically likewise well defined group (DM) for comparison:

“The aim of the study was to investigate if apoptotic myonuclei are present in muscle in the specific subgroup of IIM with partial invasion, and to detect the presence of key molecules of the apoptotic-mediating or -protective pathways in the affected and in adjacent fibres. The CD8+ T-cell-mediated immune effector mechanism and frequency of type 2 macrophages were evaluated.”

 We also clarified the rationale for selection of cases in the introduction section (line 55-61): “Another widely accepted pathology based subgroup of IIM is immune-mediated necrotizing myopathy (IMNM), also called necrotizing autoimmune myopathy (NAM) (6), characterized by muscle fibre necrosis and the relative absence of inflammatory cells. In addition, there is an ongoing discussion of further myopathological subdivisions of these diseases (7, 8). However, partial invasion is a feature that is common to both IBM and PM (strictly defined), and extensive sectioning of muscle tissue shows that partial invasion is sometimes encountered also in classical DM, as well as in overlap syndromes (1, 9). This raises the question whether there are other shared pathogenic processes, associated with partial invasion.”

1. Danielsson O, Lindvall B, Gati I, Ernerudh J. Classification and diagnostic investigation in inflammatory myopathies: a study of 99 patients. The Journal of rheumatology. 2013;40(7):1173-82.

2. Bohan A, Peter JB. Polymyositis and dermatomyositis (second of two parts). The New England journal of medicine. 1975;292(8):403-7.

3. Bohan A, Peter JB. Polymyositis and dermatomyositis (first of two parts). The New England journal of medicine. 1975;292(7):344-7.

4. van der Meulen MF, Bronner IM, Hoogendijk JE, Burger H, van Venrooij WJ, Voskuyl AE, et al. Polymyositis: an overdiagnosed entity. Neurology. 2003;61(3):316-21.

5. Benveniste O, Rider LG, Group EMOS. 213th ENMC International Workshop: Outcome measures and clinical trial readiness in idiopathic inflammatory myopathies, Heemskerk, The Netherlands, 18-20 September 2015. Neuromuscular disorders : NMD. 2016;26(8):523-34.

6. Hoogendijk JE, Amato AA, Lecky BR, Choy EH, Lundberg IE, Rose MR, et al. 119th ENMC international workshop: trial design in adult idiopathic inflammatory myopathies, with the exception of inclusion body myositis, 10-12 October 2003, Naarden, The Netherlands. Neuromuscular disorders : NMD. 2004;14(5):337-45.

7. De Bleecker JL, De Paepe B, Aronica E, de Visser M, Group EMMBS, Amato A, et al. 205th ENMC International Workshop: Pathology diagnosis of idiopathic inflammatory myopathies part II 28-30 March 2014, Naarden, The Netherlands. Neuromuscular disorders : NMD. 2015;25(3):268-72.

8. De Bleecker JL, Lundberg IE, de Visser M, Group EMMBS. 193rd ENMC International workshop Pathology diagnosis of idiopathic inflammatory myopathies 30 November - 2 December 2012, Naarden, The Netherlands. Neuromuscular disorders : NMD. 2013;23(11):945-51.

9. Arahata K, Engel AG. Monoclonal antibody analysis of mononuclear cells in myopathies. I: Quantitation of subsets according to diagnosis and sites of accumulation and demonstration and counts of muscle fibers invaded by T cells. Annals of neurology. 1984;16(2):193-208.

---

## [Decision Letter · Decision Letter 1]

16 Jul 2020

PONE-D-19-35903R1

Partial invasion and apoptosis in idiopathic inflammatory myopathies; parallel processes mediated by CD8+ cytotoxic T cells

PLOS ONE

Dear Dr. Danielsson,

Thank you for submitting your manuscript to PLOS ONE. After careful consideration, we feel that it has merit but does not fully meet PLOS ONE’s publication criteria as it currently stands. Therefore, we invite you to submit a revised version of the manuscript that addresses the points raised during the review process.

Reviewer 1 maintains that you have too few cases to draw your conclusions, and I agree with this assessment.  This is the comment from the first review, " Title of the manuscript is « Partial invasion and apoptosis in idiopathic inflammatory myopathies; parallel processes mediated by CD8+ cytotoxic T cells », nevertheless the authors have only 10 cases of IIM, including 3 IBM def (1 poss), 1 overlap def, PM….. Following the 119th ENMC classification, IIM also include IMNM and DM. Why is DM placed separately in a different group, why is there no IMNM case. Sample size is a little small to conclude." Please either evaluate more patients or explain why you think we are incorrect.

We look forward to receiving your revised manuscript.

Kind regards,

Alfred S Lewin, Ph.D.

Academic Editor

PLOS ONE

Reviewers' comments:

Reviewer's Responses to Questions

**Comments to the Author**

1. If the authors have adequately addressed your comments raised in a previous round of review and you feel that this manuscript is now acceptable for publication, you may indicate that here to bypass the “Comments to the Author” section, enter your conflict of interest statement in the “Confidential to Editor” section, and submit your "Accept" recommendation.

Reviewer #1: (No Response)

Reviewer #2: All comments have been addressed

2. Is the manuscript technically sound, and do the data support the conclusions?

Reviewer #1: (No Response)

Reviewer #2: Partly

3. Has the statistical analysis been performed appropriately and rigorously? 

Reviewer #1: (No Response)

Reviewer #2: Yes

4. Have the authors made all data underlying the findings in their manuscript fully available?

Reviewer #1: Yes

Reviewer #2: Yes

5. Is the manuscript presented in an intelligible fashion and written in standard English?

Reviewer #1: Yes

Reviewer #2: Yes

6. Review Comments to the Author

Reviewer #1: Thank you for your clear and detailed responses. Nevertheless, i still believe that sample size is small to conclude, especially that myositis are a very heterogenous group of diseases, in addition to human inter-sample heterogeneity.

Reviewer #2: (No Response)

7. PLOS authors have the option to publish the peer review history of their article (what does this mean?). If published, this will include your full peer review and any attached files.

Reviewer #1: No

Reviewer #2: No

---

## [Author Response · Author response to Decision Letter 1]

23 Aug 2020

Response to reviewers (also uploaded as a separate file)

We agree that the number of cases is rather low, and that IIM is a heterogeneous entity. We therefore propose to change the title, so that it clearly shows that we have investigated a subset of IIM, namely IIM with partial invasion, thereby reducing the heterogeneity and highlighting the focus on a particular process. In addition, we have modified the text and conclusion, to clarify that we only study the phenomenon of partial invasion (which still is highly relevant) in a subset of IIM, and that verifying studies are needed to confirm the generalizability of our findings. Unfortunately, it would be difficult and take far too long time to include mores samples to the present paper.

The following changes of the manuscript have been made:

Former title:

Partial invasion and apoptosis in idiopathic inflammatory myopathies; parallel processes mediated by CD8+ cytotoxic T cells

New title:

Apoptosis in idiopathic inflammatory myopathies with partial invasions; a role for CD8+ cytotoxic T cells?

Added in Discussion, line 333:

Further, other pathological subgroups of IIM than those with partial invasion and pathology characteristic of DM were not studied. The presence or absence of apoptosis in the IIM subgroups classified as immune-mediated necrotizing myopathy, unspecific myositis and the recently emerged group characterized by perifascicular pathology, associated with an antisynthetase syndrome [52] should be addressed in further studies.

Former Conclusion:

In conclusion: Apoptosis of muscle fibre nuclei is present in inflammatory myopathies. The affected fibres express MHC I and are surrounded by CD8+ T-cells, macrophages and granzyme B+ cells. Fibres with apoptotic nuclei occur almost exclusively in myositis associated with partial invasion, and these processes occur in the same area, but rarely in the same fibre. The findings collectively support that apoptosis, induced by cytotoxic CD8+ T-cells in IIM, may be a mechanism activated in parallel with the fibre disintegration seen in partial invasion. In addition to its apoptosis inducing potential, granzyme B may have other important roles in IIM with partial invasion. Inflammatory cells with macrophage/myeloic dendritic properties are also present in infiltrates invading muscle fibres and adhere to fibres with apoptotic myonuclei.

New Conclusion:

In conclusion: Apoptosis of muscle fibre nuclei is present in inflammatory myopathies with partial invasions, and these processes occur in the same area, but rarely in the same fibre. The affected fibres express MHC I and are surrounded by CD8+ T-cells, macrophages and granzyme B+ cells. The findings collectively support that apoptosis, induced by cytotoxic CD8+ T-cells in IIM, may be a mechanism activated in parallel with the fibre disintegration seen in partial invasion. In addition to its apoptosis inducing potential, granzyme B may have other important roles in IIM with partial invasion. Inflammatory cells with macrophage/myeloic dendritic properties are also present in infiltrates invading muscle fibres and adhere to fibres with apoptotic myonuclei. 

These text changes are red in Manuscript with Track Changes; those from the first revision are blue.

---

## [Editor Report · Decision Letter 2]

2 Sep 2020

Apoptosis in idiopathic inflammatory myopathies with partial invasion; a role for CD8+ cytotoxic T cells?

PONE-D-19-35903R2

Dear Dr. Danielsson,

We’re pleased to inform you that your manuscript has been judged scientifically suitable for publication and will be formally accepted for publication once it meets all outstanding technical requirements.

Kind regards,

Alfred S Lewin, Ph.D.

Section Editor

PLOS ONE
---

## [Editor Report · Acceptance letter]

4 Sep 2020

PONE-D-19-35903R2 

Apoptosis in idiopathic inflammatory myopathies with partial invasion; a role for CD8^+^ cytotoxic T cells? 

Dear Dr. Danielsson:

I'm pleased to inform you that your manuscript has been deemed suitable for publication in PLOS ONE. Congratulations! Your manuscript is now with our production department. 

Kind regards, 

on behalf of

Dr. Alfred S Lewin 

Section Editor

PLOS ONE